# Transpirational Leaf Cooling Effect Did Not Contribute Equally to Biomass Retention in Wheat Genotypes under High Temperature

**DOI:** 10.3390/plants11162174

**Published:** 2022-08-21

**Authors:** Helen Bramley, S. R. W. M. Chandima J. K. Ranawana, Jairo A. Palta, Katia Stefanova, Kadambot H. M. Siddique

**Affiliations:** 1The UWA Institute of Agriculture, The University of Western Australia, Perth, WA 6001, Australia; 2Plant Breeding Institute, School of Life and Environmental Sciences, The University of Sydney, Narrabri, NSW 2390, Australia; 3Department of Export Agriculture, Faculty of Animal Science and Export Agriculture, Uva Wellassa University, Badulla 90000, Sri Lanka; 4CSIRO Agriculture, Private Bag No. 5, Wembley, WA 6913, Australia

**Keywords:** abiotic stress, leaf temperature, morpho-physiological characteristics, water use

## Abstract

High temperature and water deficit are the most critical yield-limiting environmental factors for wheat in rainfed environments. It is important to understand the heat avoidance mechanisms and their associations with leaf morpho-physiological traits that allow crops to stay cool and retain high biomass under warm and dry conditions. We examined 20 morpho-physiologically diverse wheat genotypes under ambient and elevated temperatures (T_air_) to investigate whether increased water use leads to high biomass retention due to increased leaf cooling. An experiment was conducted under well-watered conditions in two partially controlled glasshouses. We measured plant transpiration (T_r_), leaf temperature (T_leaf_), vapor pressure deficit (VPD), and associated leaf morpho-physiological characteristics. High water use and leaf cooling increased biomass retention under high temperatures, but increased use did not always increase biomass retention. Some genotypes maintained biomass, irrespective of water use, possibly through mechanisms other than leaf cooling, indicating their adaptation under water shortage. Genotypic differences in leaf cooling capacity did not always correlate with T_r_ (VPD) response. In summary, the contribution of high water use or the leaf cooling effect on biomass retention under high temperature is genotype-dependent and possibly due to variations in leaf morpho-physiological traits. These findings are useful for breeding programs to develop climate resilient wheat cultivars.

## 1. Introduction

Crop productivity under the projected increased intensity and frequency of extreme climate events, including high temperature, vapor pressure deficit (VPD), and drought spells, needs genetic improvements targeting specific environmental conditions. Identifying genotypic variability in morpho-physiological responses to environmental conditions and the underlying mechanisms contributing to those variable responses are important prerequisites for genetic improvement.

Wheat is highly sensitive to heat stress [1,2], and increasing air temperature (T_air_) threatens wheat production [3,4]. High temperatures can negatively affect plant growth and development and reduce biomass accumulation leading to yield penalties [5,6,7]. Short or long-term exposure to high ambient temperatures (>35 °C) can significantly reduce wheat yield [8,9] due to accelerated senescence [10], shortened grain-filling duration, restricted carbon assimilation [11], pollen sterility [12], floret abortion [13], and infertility [14]. Tolerance to high temperatures has been widely studied and mechanisms identified, especially those supporting better photosynthetic capacity and longer grain-filling periods, such as stay green [15,16], reduced respiratory O_2_ consumption under day [17] and night warming [18], and remobilization of stem reserves [19]. Moreover, studies have demonstrated that genetic improvements in heat tolerance can be made through the incorporation of some of these traits from wild relatives of wheat [15,16]. 

Although improving heat tolerance is clearly important for wheat production, other strategies of adaptation may be just as important, but have been less explored [20,21]. For example, less stress under high temperatures may be experienced through mechanisms that help prevent overheating, called “heat avoidance” [22]. Transpiration (T_r_) is one such heat avoidance mechanism through its contribution to leaf cooling [23,24,25]. T_r_ is a complex biophysical process where water evaporates into the surrounding air through stomata. This liquid-to-vapor conversion requires energy supplied by the heat energy absorbed by the leaf [26]. During T_r_, considerable heat is dissipated from leaves allowing them to remain several degrees cooler than T_air_ [27,28,29]. Therefore, genotypes that use more water through T_r_ may avoid heat stress and be more productive under higher temperatures by maintaining cooler canopies than those that use less water. However, the role of T_r_ in regulating plant temperature is controversial. Some studies have suggested that the T_r_ contribution (latent heat loss) for regulating plant temperature is less than other heat dissipation mechanisms, especially under limited water conditions [30,31,32]. Conversely, other studies have suggested that T_r_ plays a major role in dissipating heat energy and lowering leaf temperature [29,33,34,35]. However, comparing leaf and air temperatures may underestimate the role of T_r_ in leaf cooling as the absorption coefficients differ for leaf and air, as plants have a relatively higher capacity to absorb light than air [34]. These contrasting views on the role of T_r_ in ameliorating leaf temperature necessitate closer examination of this phenomenon to determine whether this could be a suitable trait to target for wheat breeding.

Grain yield is a function of biomass and the harvest index (HI). Therefore, grain yield can be increased either by increasing the biomass, or the HI, or both [36]. A high positive correlation between grain yield and accumulated biomass was reported in several crops including rice, wheat, and maize [37,38,39]. Modern cereal breeding programs have highlighted the possibility of increasing grain yield by selecting genotypes with higher biomass production [40,41,42]. However, it is challenging to increase biomass accumulation particularly under sub-optimum growing conditions. Therefore, improvement in grain yield potential through increased biomass production can be achieved under favorable conditions, and maintaining a high HI could be more important than increasing biomass production for higher grain yield under sub-optimum conditions [43]. Biomass has been reported to decrease substantially under high temperature and the decrease in biomass under high temperature may be attributed to accelerated leaf chlorophyll degradation, reduced leaf photosynthesis, and a smaller leaf area index [44]. Moreover, variation in biomass accumulation is associated with differential soil water extraction by roots, variation in evapotranspiration (ET), and differences in net carbon assimilation per unit of water transpired (Transpiration Efficiency) or ratio of biomass to ET [45,46]. Biomass accumulation tends to be linearly related to plant T_r_ [47,48]. Thus, effective water-use via T_r_ is important for increasing biomass under water stress [49,50]. However, using more water to maintain cooler canopies and increase biomass accumulation is problematical because most wheat production occurs under rainfed conditions in water-limited environments, where water scarcity is expected to increase [51,52,53]. Therefore, it is critical to determine whether increased water use via T_r_ always contributes to high biomass accumulation and identify genotypes that can maintain cooler canopies and high biomass irrespective of water use. 

Leaf temperature (T_leaf_) has been associated with several plant traits, including stomatal conductance (g_s_) [54], capacity of vascular system to support T_r_ [55], plant water status [56], rooting depth [57], and yield [58,59]. Cooler canopies are considered to indicate better hydration status and greater capacity to extract water from deeper soil profiles [60], but a direct link between these traits has not been studied. Thus, T_leaf_ is being widely used as a selection tool for screening heat- and drought-tolerant genotypes, as it reflects genotype fitness in a given environment [55,58,61]. However, the relationship between T_leaf_ and T_r_ is complex as both traits are influenced by a multitude of morpho-physiological and environmental variables, including leaf glaucousness, canopy architecture, VPD, radiation, and soil water availability [54,55,62,63,64]. Wheat genotypes may not respond equally to elevated temperature and temperature-coupled environmental changes due to differences in their morpho-physiological attributes. Systematic characterization of the morpho-physiological differences and changes in T_r_, T_leaf_, and biomass accumulation among wheat genotypes will improve our understanding of these complex relationships. 

This study aimed to investigate the relationship between T_r_, T_leaf_, and biomass production under elevated temperature in a morpho-physiologically diverse set of 20 wheat genotypes. We aimed to determine whether increased water use under high temperature improves maintenance of shoot biomass retention (smaller reduction in biomass) in wheat due to increased leaf cooling. It was expected that elevated temperature would reduce biomass accumulation due to high temperature-induced senescence and genotypes with higher T_r_ would be cooler and thus, have slower leaf senescence and better biomass retention. It is hypothesized that biomass accumulation is less reduced in genotypes with high water use as canopy cooling inhibits leaf senescence. The study also aimed to elucidate the responses of T_r_, T_leaf_, and associated morpho-physiological characteristics in warm and dry atmospheric conditions and determine whether variable responses in the instantaneous rate of T_r_ to VPD affect T_leaf_ regulation in wheat. 

## 2. Results

### 2.1. Morpho–Physiological Responses under Elevated Temperature

#### 2.1.1. Daily T_r_

There was no genotype × temperature regime interaction for daily T_r_ per plant in Batch 01 (*p* = 0.731; Figure 1) or Batch 02 (*p* = 0.604; Figure 1). The average daily T_r_ differed significantly between genotypes in both batches (Batch 01, *p* = 0.006; Batch 02, *p* < 0.0001). 

The average daily T_r_ ranged from 94.8 g plant^−1^ d^−1^ in Wyalkatchem to 112.4 g plant^−1^ d^−1^ in Downey in Batch 01, and 117.5 g plant^−1^ d^−1^ in Kukri to 162.1 g plant^−1^ d^−1^ in Espada in Batch 02. The elevated temperature treatment (T2) significantly increased the average daily T_r_ (*p* < 0.0001) by 70.8% in Batch 01 genotypes and 38% in Batch 02 genotypes (Figure 1). However, post-hoc analysis revealed no significant difference in daily T_r_ occurred between ambient (T1) and elevated (T2) temperatures in Hartog, Kukri, and Mace (*p* > 0.05, Figure 1). 

#### 2.1.2. Leaf Glaucousness

There were noticeable differences in the presence/absence of wax and the amount of wax deposition on leaf surfaces among genotypes (Figure 2). The cultivars Downey, Einkorn, Glossy-Huguenot, and Janz had no visual signs of glaucousness, except Janz under T2, where some whitish/blue wax was deposited on the abaxial leaf surface, close to the leaf base. T2 induced whitish/blue wax deposition on the adaxial leaf surface of cultivar Magenta, relative to T1, where it only exhibited glaucousness on the abaxial side of the leaf. Drysdale, Espada, Gladius, Glennson 81, Hartog, and RAC 875 leaves were glaucous on both leaf surfaces under both temperature regimes.

Glaucousness first appeared on the flag leaf sheath and abaxial surface of the leaf lamina close to the base, and wax expression increased with time. More wax deposition was always evident on the abaxial leaf surface than the adaxial leaf surface. Based on visual observations, RAC 875, Espada, and Gladius were identified as heavy wax genotypes.

#### 2.1.3. Pubescence

Cultivar Downey was the only genotype with pubescent leaves among all genotypes, with dense hair covering both abaxial and adaxial leaf surfaces, which was easily detected with a magnifying glass.

#### 2.1.4. Stomatal Density

##### Total Stomatal Density (Abaxial and Adaxial Combined)

Batch 01 had a significant genotype × temperature regime interaction for total stomatal density (Table 1). In Batch 01, Magenta, Excalibur, and Einkorn had significantly higher stomatal density under T2 than T1 whereas all other genotypes were unaffected. Under both temperature regimes, Einkorn had the highest total stomatal density and Downey had the lowest. Batch 02 did not have a significant genotype × temperature regime interaction for total stomatal density (Table 1) but had significant main effects for genotype and temperature regime. In Batch 02, total stomatal density ranged from 101.4 mm^−2^ in Espada to 151.9 mm^−2^ in Glossy-Huguenot under T1 and increased by 11.1% (on average) under T2 relative to T1. No significant correlation occurred between total stomatal density and g_s_ (at the end of the experiment) in either batch.

##### Abaxial and Adaxial Stomatal Densities

Batch 01 had a significant genotype × temperature regime interaction for abaxial stomatal density (Table 1). The two temperature regimes produced statistically similar abaxial stomatal densities for each genotype, except for an increase in Excalibur and Einkorn and a decrease in Wyalkatchem under T2 compared with T1. Under both temperature regimes, Einkorn had the highest abaxial stomatal density and Downey had the lowest. Batch 02 had significant main effects for genotype and temperature regime, but not their interaction (Table 1), for abaxial stomatal density. In Batch 02, abaxial stomatal density ranged from 47.9 mm^−2^ in Espada to 71.9 mm^−2^ in Glossy-Huguenot under T1 and increased by 11.3% (on average) under T2 relative to T1. Unlike Batch 01, Wyalkatchem had statistically similar abaxial stomatal density under both temperature regimes in Batch 02.

Neither batch had significant genotype × temperature regime interactions for adaxial stomatal density (Table 1), with all genotypes similarly affected by temperature. In both batches, the average adaxial stomatal density significantly differed between genotypes ranging from 50.1 mm^−2^ in Downey to 72.9 mm^−2^ in Glennson 81 in Batch 01, and 57.7 mm^−2^ in Espada to 81.6 mm^−2^ in Glossy-Huguenot in Batch 02. T2 increased the average adaxial stomatal density by 12% in Batch 01 genotypes and 11% in Batch 02 genotypes, relative to T1.

### 2.2. Growth Parameters

#### 2.2.1. Green Leaf Area

Both batches had significant genotype × temperature regime interactions for green leaf area (Table 2). In general, green leaf area significantly decreased under T2 compared with T1, except for two genotypes (Sonora 64 and Downey) with similar green leaf areas between the two temperature regimes. In Batch 01 under T1, Downey had the largest green leaf area, and Sonora 64 had the smallest, whereas under T2, Downey had the largest green leaf area, and Ciano 67 had the smallest. In Batch 02, green leaf area under T1 ranged from 0.05 m^2^ plant^−1^ in Emu Rock to 0.14 m^2^ plant^−1^ in LongReach-Envoy, and under T2 ranged from 0.02 m^2^ plant^−1^ in Hartog to 0.06 m^2^ plant^−1^ in Glossy-Huguenot. 

Green leaf area per plant had a significant positive correlation with total water use per plant under both temperature regimes in both batches (Pearson’s correlation coefficient (r) = 0.4–0.6; *p* < 0.05) with comparatively stronger correlations under elevated temperature (T2).

#### 2.2.2. Specific Leaf Weight

The effect of the genotype on specific leaf weight (measure of leaf thickness) varied depending on the temperature regime in Batch 01 and Batch 02 (Table 2). In Batch 01, specific leaf weight under T1 ranged from 31.6 g m^−2^ in Einkorn to 49.3 g m^−2^ in Sonora 64 (thick leaves), and under T2 it ranged from 39.18 g m^−2^ in Downey to 52.0 g m^−2^ in Glennson 81. In Batch 02, specific leaf weight under T1 ranged from 45.9 g m^−2^ in Wyalkatchem and 56.7 g m^−2^ in Glossy-Huguenot, and under T2 it ranged from 50.2 g m^−2^ in Gladius to 57.9 g m^−2^ in LongReach-Envoy.

#### 2.2.3. Shoot Dry Weight

Only the main effects of genotype and temperature regime, but not their interaction, had a significant effect on shoot dry weight in both batches (Table 2). In Batch 01, Magenta and Janz had the highest average shoot dry weights, and Einkorn had the lowest. In Batch 02, Espada had the highest average shoot dry weight, and Wyalkatchem had the lowest. The elevated temperature decreased average shoot dry weights by approximately 20% in both batches. 

Shoot dry weight per plant had a significant positive correlation with total water use per plant under both temperature regimes in both batches (Figure 3). However, Batch 02 genotypes had a stronger correlation (Figure 3B) than Batch 01 genotypes (Figure 3A). 

#### 2.2.4. Root Dry Weight and Shoot: Root Ratio

Batch 02 did not have a significant genotype × temperature regime interaction for root dry weight but had significant main effects for genotype and temperature regime (Table 3). Glossy-Huguenot had the highest average root dry weight, whereas Emu Rock had the lowest. On average, root dry weight under T2 decreased by 42.5%, relative to T1. The effect of genotype on the shoot: root ratio varied significantly depending on the temperature regime (Table 3). The shoot: root ratio under T1 ranged from 6.2 in LongReach-Envoy to 12.3 in Emu Rock, and under T2 ranged from 6.8 in Glossy-Huguenot to 18.2 in Hartog. In general, shoot: root ratios were higher under T2 than T1. There was no significant correlation between shoot: root ratio and total water use per plant (*p* = 0.208) in either batch or temperature regime. 

### 2.3. Associations between Biomass Retention under High Temperature and Total Water Use Per Plant and Leaf-to-Air Temperature Differential (T_air_–T_leaf_)

Table 4 summarizes the results for the fixed effects in the linear mixed model, where biomass reduction% under T2 relative to T1 was fitted as the response variable. The (T_air_–T_leaf_) × genotype and total water use × genotype interactions were highly significant (Table 4), indicating that the effects of leaf-to-air temperature differential and total water use per plant on biomass reduction% under high temperature depends on the genotype. 

Biomass reduction% had a significant negative correlation with total water use per plant in both batches. However, Batch 02 had a stronger correlation (r = −0.52; *p* < 0.001) than Batch 01 (r = −0.36; *p* = 0.021).

### 2.4. Genotypic Variability in T_leaf_ Response to T_air_

T_leaf_ had a positive linear correlation with T_air_ for all genotypes, but the increase in T_leaf_ per unit increase in T_air_ (i.e., slope) significantly differed between genotypes (Table 5). The T_leaf_ (T_air_) regression slope ranged from 0.49 °C °C^−1^ in Einkorn to 1.02 °C °C^−1^ in Janz. 

### 2.5. Relationship between Instantaneous T_r_ Response to VPD and T_leaf_ Regulation

The instantaneous T_r_ increased linearly with VPD, but the response (i.e., slope of the regression) varied among genotypes (*p* < 0.05), ranging from 1.04 mmol H_2_O m^−2^ s^−1^ kPa^−1^ in Hartog to 3.08 mmol H_2_O m^−2^ s^−1^ kPa^−1^ in Magenta (Appendix A). Figure 4 shows the relationship between T_leaf_ (T_air_) response and instantaneous T_r_ (VPD) response in the 20 wheat genotypes studied. The T_leaf_ response to T_air_ had no significant correlation with T_r_ response to VPD when all 20 wheat genotypes were considered together (*p* = 0.981). Several genotypes did not conform to the general trend, with T_leaf_ of Janz, Magenta, and Glennson 81 (Figure 4) increasing in parallel with T_air_, but their instantaneous T_r_ increased rapidly with VPD (greater slope of instantaneous T_r_ vs. VPD regression). In contrast, T_leaf_ of Espada and Kukri increased slowly with T_air_, but their instantaneous T_r_ increased more slowly with VPD. After removing these five contrasting genotypes, the T_leaf_ response to T_air_ had a significant negative correlation with the instantaneous T_r_ response to VPD (r = −0.60; *p* = 0.013). 

## 3. Discussion

High temperature and high VPD increased the daily T_r_ and total water use in both batches of genotypes. The effect of water-use and the leaf-to-air temperature differential (T_air_–T_leaf_) on shoot biomass retention under high temperature depended on the genotype due to their different morpho-physiological and growth attributes. Traits such as stomatal density, glaucousness, green leaf area, and shoot and root dry weights were also affected by high temperature and VPD. Instantaneous T_r_ increased with VPD in all genotypes, but the rate of increase (slope of the relationship between instantaneous T_r_ and VPD) differed between genotypes, indicating genotypic variability in this trait. Genotypic variability in the T_leaf_ response to T_air_ was apparent. In some genotypes (e.g., Einkorn, RAC 875, Excalibur), T_leaf_ increased slowly with T_air_, whereas in others (e.g., Janz, Sonora 64, Magenta) T_leaf_ increased rapidly with T_air_. In most genotypes, the T_leaf_ response to T_air_ was correlated with the instantaneous T_r_ response to VPD, indicating that genotypes whose instantaneous T_r_ increased more with VPD maintained lower T_leaf_ as T_air_ increased and vice versa. However, for several contrasting genotypes, the T_leaf_ response to T_air_ did not correlate with the instantaneous T_r_ response to VPD.

### 3.1. Greater Water Use through T_r_ Increases Shoot Biomass Retention under High Temperature

Under high temperatures, plant survival and productivity rely on heat tolerance and avoidance mechanisms [65]. Heat avoidance is the ability to maintain internal tissue temperatures below lethal stress levels, which is acquired through transpirational cooling, changes in leaf orientation, solar radiation reflection, and extensive rooting [66,67]. In contrast, heat-tolerant genotypes survive and develop under high temperatures, even with high internal tissue temperatures [67]. Increased biomass retention under high temperatures has been associated with heat tolerance and avoidance mechanisms. This study showed that high water use contributes to increased biomass retention under high temperatures depending on the genotype. CO_2_ exchange and water escape share a common pathway through the stomata. Thus, this contribution of greater water use to biomass retention under high temperatures may be due to greater CO_2_ exchange and evaporative cooling. 

Higher water use by some genotypes (e.g., Magenta and Janz) did not always result in greater biomass retention. Biomass retention in those genotypes could be regulated by photosynthetic efficiency, biomass accumulation, and respiration. Some genotypes may be more efficient at photosynthesis and biomass accumulation and less sensitive to heat, despite differences in water use. The effect of leaf cooling or leaf-to-air temperature differential on biomass retention under high temperature also varied between genotypes. In some genotypes, biomass retention was not associated with the leaf-to-air temperature differential, suggesting that biomass retention under high temperature may not be linked to water use or its resultant effect on leaf cooling (see Appendix A). 

Other possible reasons why T_air_–T_leaf_ was not directly related to biomass retention under high temperature include:(1)Environmental factors that may confound the relationship between T_air_–T_leaf_ and T_r_, such as VPD, T_air_, net radiation, and boundary layer resistance [68]. For instance, T_air_–T_leaf_ may not be constant and may increase with T_air_. Ideally, T_leaf_ measurements in each genotype should be made at the same T_air_ to prevent differences in T_air_ confounding the results.(2)Biomass retention under high temperature may be related to heat tolerance, independent of T_air_–T_leaf_ or T_r_. The internal tolerance mechanisms of some genotypes could enable them to withstand the heat, despite high T_leaf_, without reducing biomass. For instance, the production of heat shock proteins that have a protective role against heat stress can be induced in some wheat genotypes exposed to temperatures around 32 °C [69]. In this study, T_air_ sometimes exceeded 35 °C, suggesting that some genotypes acquired heat tolerance. However, the reverse can also happen, where some genotypes become more heat-sensitive, even with relatively low T_leaf_.(3)T_air_–T_leaf_ measured once or at different times during the experimental period may not relate to biomass retention or water use during the whole period. The ability to maintain greater T_air_–T_leaf_ throughout the plant’s growing cycle, regardless of growth stage or changes in environmental conditions, is more important (stability) and translates into biomass retention under warmer conditions. This trait may be more prominent in some genotypes than others.

### 3.2. Water Use through T_r_ and Associated Morpho-Physiological and Growth Characteristics as Affected by Temperature 

High temperatures (8–9 °C rise in average temperature) increased the average daily T_r_ by 38–71% depending on the batch. Such increases in daily T_r_ are likely to be driven by the associated higher VPD under high temperature [62,70] and the changes in morpho-physiological characteristics induced by high temperature and VPD. Interestingly, the daily T_r_ for three genotypes with relatively low daily T_r_ (Hartog, Kukri, and Mace) did not significantly differ between the two temperature regimes and may have strict control of water use through stomata or other hydraulic properties.

Overall, high temperatures significantly affected morpho-physiological and growth characteristics, including stomatal density, amount of leaf glaucousness, green leaf area, and shoot and root biomass, but the amount or direction of the changes varied between genotypes. 

The observed increase in daily T_r_ under high temperature may be related to changes in stomatal density. Abaxial and adaxial stomatal density varied between genotypes and between temperature regimes. For example, stomatal density in wheat decreased in the flag leaf but increased in the penultimate leaf with increasing temperature [71]. Stomatal density also increased in a range of herbs and trees in response to high temperature and VPD [72,73,74], but decreased under high VPD in some species such as tomato, capsicum, and eggplant [75]. In Batch 01, total stomatal density increased in three of the ten genotypes under high temperature but was similar under both temperature regimes for the other seven genotypes. In Batch 02, total stomatal density increased by 11.1% for all genotypes under high temperature relative to ambient conditions. This increase occurred either by the presence of more stomata on both sides of the leaves (Excalibur, Einkorn, Espada, Mace, Gladius, and Hartog) or only the abaxial (Drysdale and Glossy-Huguenot) or adaxial (Magenta) side. Positive correlations between stomatal density and g_s_ have been reported in various tree, shrub, and herbaceous species [76], but not in rice [77]. We also found no correlation between stomatal density and g_s_, which may be due to other factors affecting g_s_, such as stomatal aperture [78]. 

High temperatures increased the level of wax expression in most but not all genotypes. Although the presence of wax could reduce T_r_ due to its low permeability and blockage of the stomata [79], no such differences in daily T_r_ were observed in this study. This is likely due to other genotypic factors confounding the influence of glaucousness on daily T_r_. Consequently, precise identification of the effects of glaucousness on daily T_r_ should involve isogenic lines with a contrasting presence/absence of wax [63]. Chemical analyses are needed to estimate the level of wax deposition.

Green leaf area significantly decreased under high temperature, the extent of which depended on genotype. There was a moderate positive correlation (r = 0.40–0.57) between total water use through T_r_ and green leaf area (increased transpirational area). Genotypes with high water use might maintain leaf area by avoiding overheating and delaying senescence, which would support biomass accumulation due to greater photosynthetic area, translating into greater yields.

In both batches, high temperature (T2) reduced shoot dry weight by 20%. However, shoot dry weight positively correlated with total water use under both temperature regimes, reflecting T_r_’s positive role in plant growth and biomass accumulation. High temperatures reduced root dry weight more than shoot dry weight, which is likely associated with reduced dry matter partitioning to roots under high temperature [80]. However, the smaller root biomass met the increased evaporative demand, as indicated by a 71% greater daily T_r_ under high temperature. Root length was not measured in this study, so it is unknown whether the change in root biomass resulted in a smaller root system or thinner roots. In this scenario, water absorption and delivery (hydraulic conductance) may have increased to support the higher daily T_r._


The effect of high temperature on specific leaf weight, an approximate measure of leaf thickness, depended on genotype. In 13 of the 20 genotypes, high temperature increased specific leaf weight (leaves thickened), whereas the other 7 had similar or slightly lower specific leaf weights. A similar increase in specific leaf weight when T_air_ > 20 °C was reported previously [81]. Thick leaves are believed to have a higher volume of vascular tissues [82], improving axial hydraulic conductance. Thus, it was expected that genotypes with thick leaves would transpire more water. However, thick leaves may have a longer radial path through mesophyll, limiting radial hydraulic conductance. These counteracting effects could explain why total water use did not correlate with specific leaf weight. 

### 3.3. Different Responses between the Two Batches of Genotypes as Explained by Differential Adaptations to Warm and Dry Conditions

Variability in the performance of Batch 01 and Batch 02 genotypes highlights their sensitivity to changes in temperature and VPD. Such responses can be expected under field conditions, which are highly variable and predicted to increase. The growing conditions before exposure to high temperatures differed slightly between the two batches. Due to starting later in the season, the second batch of plants was grown under relatively warmer and drier atmospheric conditions before exposure to T2. Consequently, the second batch of genotypes may have developed some adaptive mechanisms to those warmer and drier atmospheric conditions that reduced the response to T2, relative to the first batch of genotypes. For example, VPD between the two temperature glasshouses ranged from 1.2 to 2.2 kPa (average 1.7 kPa) for Batch 02 and 0.8 to 2.2 kPa (average 1.5 kPa) for Batch 01. However, the average increase in daily T_r_ under high temperature, relative to ambient temperature, was about 38% for Batch 02 genotypes and 71% for Batch 01 genotypes. The genotype Wyalkatchem, included in both batches as a reference, transpired at a 56% higher rate under high temperature than ambient temperature in Batch 02, much lower than Batch 01 (79%). 

Furthermore, differential adaptations to warm and dry conditions in Batch 01 and Batch 02 genotypes were evident in the strength of the relationships between total water use and some growth attributes. For instance, Batch 02 had a stronger relationship between total water use and green leaf area and shoot dry weight than Batch 01. Therefore, we can assume that the influence of global warming on water use will differ between wheat grown in cooler and warmer regions. If wheat crops grown in cooler regions are exposed to warmer conditions, their water use may significantly increase, but wheat crops already in warmer conditions may not increase their water use proportionately in response to temperature.

The study suggests that some genotypes may adapt and develop tolerance mechanisms when exposed to warmer conditions for long periods. Therefore, we can expect that wheat crops already in warmer conditions may not increase their water use proportionately in response to temperature. However, if wheat crops grown in cooler regions are exposed to warmer conditions due to global warming, their water use may significantly increase. 

### 3.4. Association of Instantaneous T_r_ Response to VPD and T_leaf_ Regulation

The instantaneous rate of T_r_ linearly increased with VPD in all genotypes, but the magnitude of the increase per unit change in VPD (i.e., slope) differed between genotypes. Such variability in the instantaneous T_r_ response to VPD may be associated with differences in stomatal regulation and/or the efficacy of water absorption and delivery of the vascular system in roots, shoots, and leaves (hydraulic conductance). 

T_leaf_ also increased in response to T_air_ in all genotypes. However, the rate of increase (slope) differed among genotypes indicating differences in their capacity to regulate T_leaf_ under changing T_air_. When all 20 genotypes were considered together, no significant correlation was observed between the instantaneous T_r_ response to VPD and T_leaf_ response to T_air._ However, a positive correlation occurred when several contrasting genotypes were excluded from the analysis. The contrasting responses could be linked to differences in leaf physical attributes, such as glaucousness, pubescence, leaf color, and leaf size, affecting the absorption of radiant energy, reflectance, and sensible heat loss [83,84]. For instance, high leaf glaucousness in Espada [85] could explain its moderate T_leaf_ response to T_air_ despite its low instantaneous T_r_ response to VPD due to greater reflectance associated with glaucousness [84]. Leaf color and chlorophyll content could also affect leaf temperature [27,35,86]. Darker leaves absorb more radiation heat than pale leaves [87]. Therefore, dark green leaves may be warmer (e.g., Magenta), regardless of the T_r_ response to VPD. Traits like leaf size can affect transpirational cooling and sensible heat loss. When water is not limited, transpirational cooling is effective for large leaves due to the greater surface area for water escape. However, when water is scarce—imposing limits on transpirational cooling—large leaves may be disadvantaged due to their thick boundary layer, preventing sensible heat loss [64]. These findings suggest that transpirational cooling is not the only mechanism contributing to heat avoidance under high temperature; other heat dissipation mechanisms, such as those mentioned above, can contribute to heat avoidance. Moreover, their relative contribution varies depending on genotype.

## 4. Materials and Methods

### 4.1. Plant Material and Growing Conditions

Twenty wheat (*Triticum aestivum* L.) genotypes with contrasting morpho-physiological attributes and responses to heat and drought were selected (Table 6). The study was conducted in two batches of ten genotypes (Table 6), planted ten weeks apart due to the many measurements. The cultivar Wyalkatchem, a high-yielding and putatively drought-tolerant commercial cultivar, was used in both batches as a reference for comparison. The experiment was conducted in naturally-lit glasshouses with a daily maximum PAR 800–1000 µmol m^−2^ s^−1^ measured at the plane of the top leaves, at The University of Western Australia, Crawley, Western Australia (31°98′ S, 115°82′ E) under partially regulated conditions.

Seeds were surface sterilized and pre-germinated as described by Bramley et al. [104]. Three days after germination, seedlings were planted into the same soil and pots described in our previous study [70]. The pH (in 0.01 M CaCl_2_) and electrical conductance of the soil and sand mixture were 7.8 and 12.6 mS m^−1^, respectively. Before planting, the pot soil water capacity (PSWC) was determined by weighing each pot after saturating the soil with water and allowing them to drain freely for 48 h. The pots were sealed at the top with aluminum foil during free drainage to prevent evaporation from the soil surface. At planting, each pot was fertilized with 4 g of commercial slow-release fertilizer (N:P:K 19.4:1.6:5, Osmocote™, Scotts Australia Pty Ltd., Auckland, New Zealand). The pots were well-watered and maintained at 100 g below PSWC (85% PSWC) throughout the initial growing period by weighing daily to prevent anaerobic conditions. A liquid fertilizer (N:P:K 19:8.4:15.8, Poly–Feed™, Haifa Chemicals, Haifa, Israel) was applied (2 g per pot in 125 mL water) three weeks after planting. A 20 mm layer of white plastic beads covered the soil surface of each pot to prevent soil evaporation [70,105,106,107]. The pots were randomly arranged on a glasshouse bench and periodically rotated to minimize environmental heterogeneity. 

### 4.2. Temperature Treatment

Seven weeks after sowing when the flag leaf ligule was visible (Z39; Zadoks’ growth scale) [108], a high-temperature treatment was imposed by transferring half of the plants (four replicates of each genotype) to another glasshouse with average temperatures 8–9 °C warmer (T2) than the ambient temperature glasshouse (T1) (see Appendix A). The remaining plants were maintained in the ambient glasshouse (T1). Plants were evaluated under the two temperature regimes for four weeks.

### 4.3. Experimental Design

The experiment was a completely randomized, two-factor factorial design with four replicates per treatment combination. The two factors were genotype and temperature regime [ambient (T1) and high (T2)].

### 4.4. Measurements 

#### 4.4.1. Growth and Morpho–Physiological Parameters

The transpiration rate (T_r_) per plant was recorded by weighing pots daily between 08:00 and 09:00 h using an electronic balance with 0.1 g resolution (Model PGL 6001, Adam Equipment Company, Oxford, CT, USA). The difference between daily pot weights was considered the daily T_r_ per plant. After weighing, plants were re-watered to 100 g below PSWC. Leaf temperature (T_leaf_) was recorded on the most recent, fully-expanded sunlit leaf (middle portion of the leaf) on the tagged main stem between 10:30 and 14:30 h weekly, on clear sunny days using a hand-held infrared thermometer (Impac Model IN 15 plus, LumaSense Technologies, Santa Clara, CA, USA; ±0.2 °C precision). The emissivity of the infrared thermometer was set at 0.94 (determined in a preliminary experiment), where the T_leaf_ reading from the infrared thermometer was compared against the T_leaf_ measured with a thermocouple (Digi–Temper Model 3527 with type K thermocouples, Tsuruga Electric Works, Osaka, Japan). T_air_ surrounding the leaf was measured using a digital thermometer probe (Dick Smith Electronics Ltd., New South Wales, Australia) shaded with an aluminum sheet to prevent radiant heating from direct sunlight. The instantaneous leaf-level T_r_ and g_s_ were measured weekly on the same leaf after T_leaf_ measurements using a portable gas exchange system (Model LI-6400, LI-COR Inc., Lincoln, NE, USA). During the measurements, the conditions in the leaf chamber were: reference CO_2_ concentration 380 µmol mol^−1^, flow rate 400 µmol s^−1^, and photosynthetic photon flux density 1000 µmol m^−2^ s^−1^, which is the average saturation light intensity for photosynthesis in wheat [109]. Measurements continued for four weeks after applying the temperature treatment.

At the end of the experiment, the leaves used for physiological measurements were kept separate in zip-lock bags on ice to analyze abaxial and adaxial stomatal density. The number of stomata was counted with ImageJ 1.47v (National Institute of Health, Bethesda, Maryland, USA) from images taken using a digital camera (Pro-MicroScan Model DCM 310, Oplenic Optronics Co., Ltd., Hangzhou, China) mounted on a dissecting microscope (Zeiss SteREO Discovery.V8, Carl Zeiss MicroImaging GmbH, Gottingen, Germany). Leaf glaucousness was observed on the adaxial and abaxial surfaces of the leaf lamina (same leaf used for gas exchange measurements) using a magnifying glass (3×) and scored using a 0–3 scale, based on visual observations (0 = no wax, 1 = wax only on adaxial surface, 2 = wax only on abaxial surface, and 3 = wax on both adaxial and abaxial surfaces). Similarly, leaf pubescence was rated on a 0–3 scale (0 = hairs, 1 = hairs on both adaxial and abaxial surfaces, 2 = hairs only on adaxial surface, 3 = hairs only on abaxial surface). At the end of the experiment, the green leaf area of the whole plant was measured with a leaf area meter (Model LI 3000, Li-COR Inc., Lincoln, NE, USA). Leaves and stems were then oven-dried at 60 °C for 48 h for dry weight measurements. The specific leaf weight (g m^−2^) of the leaves used for gas exchange measurements was calculated as the ratio of leaf dry weight to leaf area. Roots were carefully washed from the soil as described by Liao et al. [110] and Palta et al. [111], and dry weight recorded after oven drying at 60 °C for 72 h. The percentage reduction in shoot dry weight (biomass reduction%) under elevated temperature (T2) in relation to ambient temperature (T1) was calculated as:$$\mathrm{Biomass}\;\mathrm{reduction}\%=\frac{\left(\mathrm{shoot}\;\mathrm{dry}\;\mathrm{weight}\;\mathrm{at}\;T1-\mathrm{shoot}\;\mathrm{dry}\;\mathrm{weight}\;\mathrm{at}\;T2\right)}{\left(\mathrm{shoot}\;\mathrm{dry}\;\mathrm{weight}\;\mathrm{at}\;T1\right)}\times 100$$

#### 4.4.2. Environmental Parameters 

Maximum and minimum T_air_ and relative humidity (RH) in the glasshouses were recorded throughout the experimental period using a data logger (WatchDog Micro Stations, Model 1250, Spectrum Technologies, Inc., Aurora, IL, USA) positioned just above the canopy. Atmospheric VPD was calculated according to the equations of [112]:$$\mathrm{VPD}=\left(\frac{100-\mathrm{RH}}{100}\right)\times \;\mathrm{SVP}\;$$
$$\mathrm{SVP}=610.7\times 10^{7.5T/\left(237.3+T\right)}$$
where RH is relative humidity (%), SVP is saturated vapor pressure (Pa), and T is air temperature (°C). 

### 4.5. Statistical Analysis

A linear mixed model was fitted for biomass reduction% to test the hypothesis that high water use maintains biomass accumulation (less reduction) under high temperature through increased leaf cooling. The blocking structure that accounted for the nested genotype structure within a batch, namely the terms batch and batch genotype, were fitted as random factors. The model accounted for the unbalanced genotype allocation within batches. The fixed terms in the model included total water use, leaf-to-air temperature differential (T_air_–T_leaf_), and their interactions with genotype. The correlation between biomass reduction% and total water use per plant was studied to identify how water use influences biomass retention under warm and dry conditions. 

Two-way ANOVAs were performed for daily T_r_, stomatal density, green leaf area, shoot and root dry weights, shoot-to-root ratio, and specific leaf weight. Treatment means were compared using the 5% least significant difference (LSD) values. Correlations were examined between total water use per plant and green leaf area, shoot dry weight, shoot-to-root ratio, and specific leaf weight. The correlation between total stomatal density and g_s_ was also examined. 

Genotypic variation in the instantaneous T_r_ response to naturally fluctuating atmospheric VPD and the T_leaf_ response to T_air_ in the glasshouse were evaluated using the single linear regression model. Data within a typical VPD range (1.3–4.3 kPa) and T_air_ range (23.6–32.3 °C) were selected for each genotype in both batches of the instantaneous T_r_ (VPD) and T_leaf_ (T_air_) regressions, respectively. Genotypes contrasting in their T_r_ response to VPD and T_leaf_ response to T_air_ were identified based on their regression slopes. The possible role of T_r_ in regulating T_leaf_ was identified by evaluating the correlation between the instantaneous T_r_ response to VPD and T_leaf_ response to T_air_. 

Statistical analysis was conducted using the R 3.6.2 programming environment [113], ASReml–R 3.0 [114], and GraphPad Prism 6.04 (GraphPad Software Inc., La Jolla, CA, USA; www.graphpad.com) (accessed on 10 May 2022).

## 5. Conclusions

High temperature and VPD significantly affected daily T_r_, T_leaf_, and associated morpho-physiological and growth characteristics. A positive correlation between shoot dry weight and water use indicates the benefit of T_r_ in maintaining growth under warm conditions. However, high water use through T_r_ or its potential effect of leaf cooling did not contribute equally to biomass retention under warm conditions in all genotypes. Some genotypes could maintain growth and biomass, to a certain extent, regardless of water use or leaf cooling through other mechanisms. Such genotypes may adapt and perform better when leaf cooling through T_r_ is limited by soil water deficit. Moreover, genotypes capable of reflecting excessive radiation heat loads (e.g., waxy or pubescent genotypes), such as Gladius and Espada, will be beneficial in such situations.

The plant root system plays a crucial role in supporting T_r_ through water absorption. It appears that when less dry matter was allocated to roots under high temperature, it did not significantly affect T_r_. Thus, root water absorption (greater hydraulic conductance) may have improved to support T_r_. However, increased dry matter allocation to produce extensive and deeper root systems under water stress would be beneficial for capturing water in deeper soil layers. 

We believe that these findings will provide new insight into the basis of selecting superior wheat genotypes for future climate scenarios. Relying only on T_leaf_ or water use measurement for screening genotypes may not be sufficient as those parameters do not always correlate with heat tolerance and productivity. As highlighted in our study, comprehensive examination of different morpho-physiological attributes is needed for screening suitable genotypes for specific environments. Genotypic performances identified in this study will be useful in breeding programs of wheat. 

## Figures and Tables

**Figure 1 plants-11-02174-f001:**
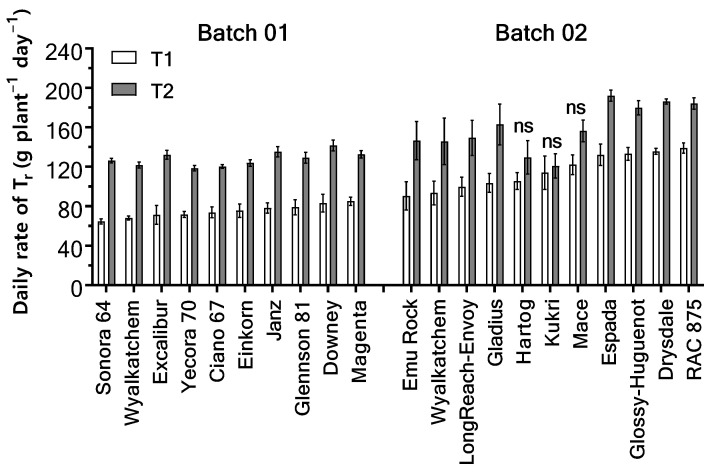
Mean daily rate of transpiration (T_r_) of 20 wheat genotypes in Batch 01 and 02 under ambient (T1) and high (T2) temperature regimes over the period of temperature treatment. Mean daily rate of T_r_ was calculated by dividing total water use per plant over the treatment period by the duration of the temperature treatment. Error bars indicate ± SEM; *n* = 4. The letters “ns” indicate no significant difference (*p* > 0.05) in the average daily rate of T_r_ between the two temperature regimes.

**Figure 2 plants-11-02174-f002:**
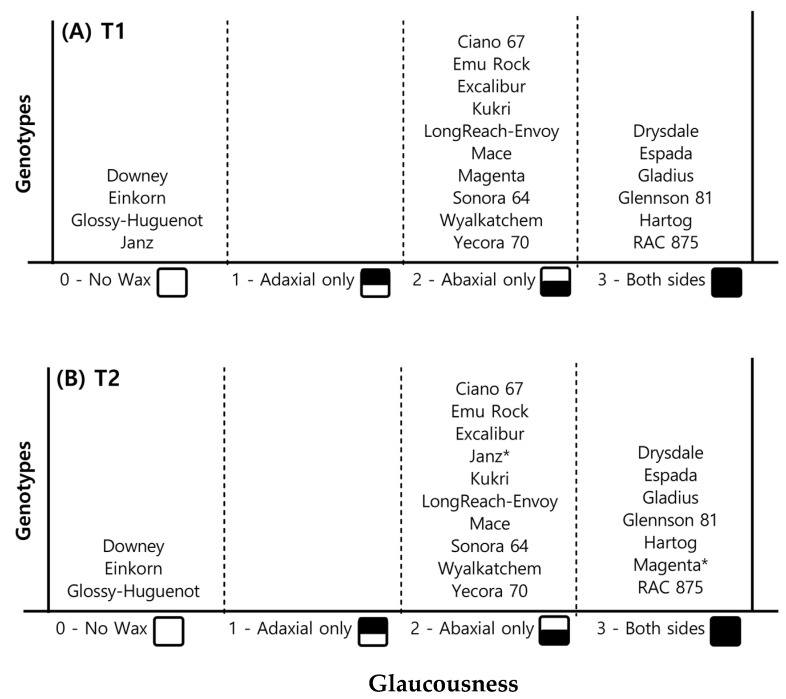
Categorization of wheat genotypes (Batch 01 and Batch 02) based on glaucousness (blueish/white waxy deposition) of leaves under (**A**) ambient (T1), and (**B**) high (T2) temperature regimes. T2 enhanced glaucousness, especially in genotypes indicated by *.

**Figure 3 plants-11-02174-f003:**
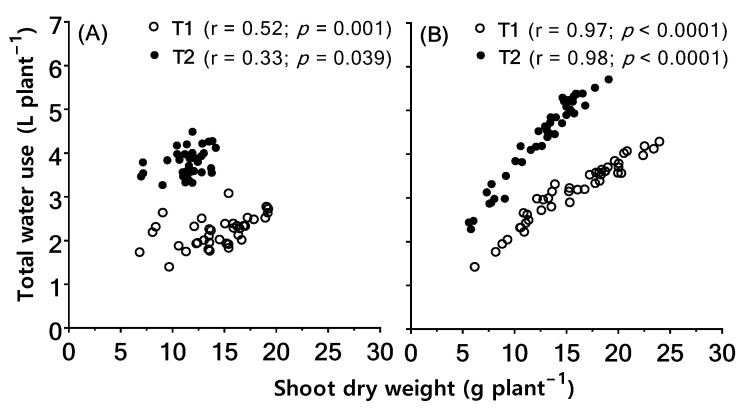
Relationship between shoot dry weight and total water use per plant during the period of temperature treatment in Batch 01 (**A**) and Batch 02 (**B**) wheat genotypes under T1 and T2 temperature conditions. Each data point represents an individual plant (replicate), all genotypes combined. Pearson correlation coefficient (r) and probability (*p*) are given in each sub-figure.

**Figure 4 plants-11-02174-f004:**
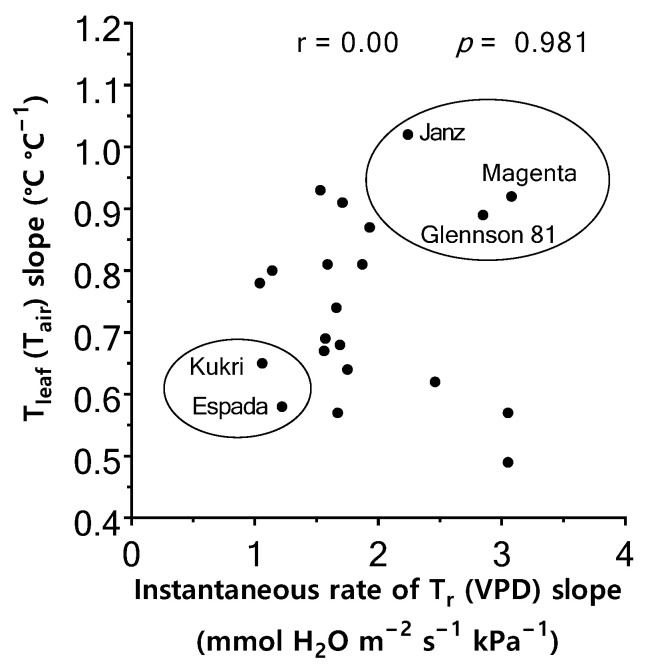
Relationship between instantaneous rate of transpiration (T_r_) response to VPD (mean slope of instantaneous rate of T_r_ response to VPD regression) and leaf temperature (T_leaf_) response to air temperature (T_air_) (mean slope of T_leaf_ response to T_air_ regression) of 20 wheat genotypes (two batches combined). Each data point represents an individual genotype. The genotypes circled do not conform to the expected model of negative linear correlation between T_leaf_ (T_air_) and instantaneous rate of T_r_ (VPD) slopes.

**Table 1 plants-11-02174-t001:** Abaxial, adaxial, and total stomatal densities of 20 wheat genotypes in Batch 01 and 02 under ambient (T1) and high (T2) temperature regimes. Values presented are means ± SEM, *n* = 4.

Batch 01					Batch 02				
Genotype (G)	Treatment (T)	Stomatal Density (Number of Stomata per mm^−2^)			Genotype (G)	Treatment (T)	Stomatal Density (Number of Stomata per mm^−2^)		
		Abaxial	Adaxial	Total			Abaxial	Adaxial	Total
Excalibur	T1	58.7 ± 1.7	59.1 ± 1.1	117.9 ± 2.0	Drysdale	T1	49.1 ± 1.4	61.1 ± 1.5	110.2 ± 2.5
	T2	71.2 ± 1.2	70.7 ± 5.2	141.8 ± 6.3		T2	58.5 ± 1.1	67.0 ± 1.3	125.5 ± 1.7
Glennson 81	T1	52.5 ± 2.4	66.7 ± 3.5	119.2 ± 1.3	LongReach-Envoy	T1	60.7 ± 1.8	61.3 ± 2.6	122.1 ± 2.4
	T2	53.0 ± 3.5	79.2 ± 4.4	132.2 ± 6.9		T2	63.7 ± 2.2	70.1 ± 2.5	133.9 ± 2.4
Sonora 64	T1	55.7 ± 2.8	64.0 ± 3.6	119.7 ± 5.3	Hartog	T1	57.7 ± 0.6	64.2 ± 1.7	122.0 ± 1.3
	T2	55.9 ± 2.5	63.0 ± 2.4	118.9 ± 1.3		T2	66.9 ± 0.9	78.3 ± 1.6	145.2 ± 2.4
Downey	T1	36.5 ± 2.4	46.8 ± 6.1	83.3 ± 8.3	Glossy-Huguenot	T1	71.9 ± 1.3	80.0 ± 1.0	151.9 ± 2.1
	T2	43.0 ± 1.0	52.5 ± 4.3	95.5 ± 5.0		T2	82.7 ± 3.3	83.1 ± 2.3	165.9 ± 4.2
Einkorn	T1	66.0 ± 5.5	59.2 ± 2.3	125.2 ± 7.7	RAC 875	T1	49.0 ± 5.8	62.2 ± 4.3	111.2 ± 10.0
	T2	80.5 ± 3.0	73.5 ± 4.9	154.0 ± 7.6		T2	55.4 ± 1.2	66.9 ± 2.3	122.2 ± 2.0
Ciano 67	T1	51.3 ± 2.0	53.3 ± 2.2	104.7 ± 4.1	Espada	T1	47.9 ± 2.9	53.5 ± 2.4	101.4 ± 4.1
	T2	54.8 ± 3.2	56.5 ± 2.7	111.3 ± 5.9		T2	57.5 ± 5.1	62.0 ± 6.1	119.5 ± 11.2
Yecora 70	T1	52.3 ± 4.1	70.2 ± 0.4	122.5 ± 4.1	Mace	T1	50.0 ± 0.6	52.0 ± 0.3	102.0 ± 0.7
	T2	60.0 ± 2.5	63.2 ± 1.2	123.2 ± 3.7		T2	57.9 ± 5.6	66.0 ± 1.8	123.9 ± 6.5
Magenta	T1	46.5 ± 3.2	60.0 ± 3.1	106.5 ± 2.5	Gladius	T1	53.2 ± 1.9	65.9 ± 2.2	119.1 ± 4.1
	T2	53.1 ± 4.9	72.6 ± 3.7	125.7 ± 8.1		T2	64.0 ± 4.2	74.2 ± 4.1	138.2 ± 5.6
Janz	T1	49.2 ± 2.1	56.2 ± 1.7	105.5 ± 2.4	Kukri	T1	54.5 ± 4.3	62.2 ± 1.6	116.7 ± 5.4
	T2	51.1 ± 3.8	67.0 ± 2.1	118.1 ± 5.6		T2	54.5 ± 3.3	62.8 ± 2.1	117.3 ± 5.2
Wyalkatchem	T1	65.4 ± 1.4	55.6 ± 0.9	121.0 ± 2.0	Emu Rock	T1	61.9 ± 1.9	63.5 ± 1.5	125.4 ± 1.5
	T2	53.4 ± 1.2	65.0 ± 2.0	118.4 ± 2.5		T2	63.4 ± 3.4	65.9 ± 3.7	129.2 ± 5.3
					Wyalkatchem	T1	59.4 ± 1.8	63.4 ± 4.5	122.7 ± 5.8
					(Reference)	T2	60.5 ± 1.9	68.6 ± 3.6	129.1 ± 5.1
*p* value: G		<0.001	<0.001	<0.001			<0.001	<0.001	<0.001
T		0.008	<0.001	<0.001			<0.001	<0.001	<0.001
G × T		0.003	0.056	0.037			0.502	0.309	0.392
LSD value: G		6.010	6.508	10.194			5.960	5.669	9.861
T		2.655	2.875	4.503			2.539	2.415	4.200
G × T		8.540	NS	14.486			NS	NS	NS

**Table 2 plants-11-02174-t002:** Green leaf area, shoot dry weight, and specific leaf weight of 20 wheat genotypes in Batch 01 and Batch 02 under ambient (T1) and high (T2) temperature regimes. Values presented are means ± SEM, *n* = 4.

Batch 01					Batch 02				
Genotype (G)	Treatment (T)	Green Leaf Area (m^2^ plant^−1^)	Shoot Dry Weight (g plant^−1^)	Specific Leaf Weight (g m^−2^)	Genotype (G)	Treatment (T)	Green Leaf Area (m^2^ plant^−1^)	Shoot Dry Weight (g plant^−1^)	Specific Leaf Weight (g m^−2^)
Excalibur	T1	0.14 ± 0.03	13.9 ± 1.9	42.7 ± 1.3	Drysdale	T1	0.08 ± 0.00	19.3 ± 0.6	52.5 ± 1.0
	T2	0.07 ± 0.01	11.7 ± 0.4	49.6 ± 0.8		T2	0.03 ± 0.00	15.0 ± 0.5	51.4 ± 2.2
Glennson 81	T1	0.17 ± 0.01	15.3 ± 1.6	46.8 ± 1.8	LongReach-Envoy	T1	0.14 ± 0.01	12.5 ± 1.2	45.9 ± 2.2
	T2	0.07 ± 0.01	10.6 ± 0.6	51.9 ± 1.4		T2	0.05 ± 0.00	12.0 ± 1.3	57.9 ± 2.7
Sonora 64	T1	0.09 ± 0.00	14.6 ± 0.7	49.3 ± 1.9	Hartog	T1	0.07 ± 0.01	14.3 ± 1.6	50.0 ± 1.5
	T2	0.06 ± 0.00	12.7 ± 0.6	47.9 ± 0.8		T2	0.02 ± 0.00	9.5 ± 1.7	51.6 ± 1.1
Downey	T1	0.24 ± 0.02	12.7 ± 1.0	34.7 ± 3.1	Glossy-Huguenot	T1	0.11 ± 0.00	19.5 ± 1.3	56.7 ± 2.6
	T2	0.23 ± 0.01	10.8 ± 0.5	39.2 ± 2.8		T2	0.06 ± 0.00	14.0 ± 0.7	57.0 ± 1.1
Einkorn	T1	0.20 ± 0.00	8.1 ± 0.5	31.6 ± 2.2	RAC 875	T1	0.10 ± 0.00	20.6 ± 1.2	54.6 ± 1.7
	T2	0.12 ± 0.00	7.2 ± 0.2	40.8 ± 1.7		T2	0.05 ± 0.00	15.4 ± 0.3	57.3 ± 1.8
Ciano 67	T1	0.09 ± 0.00	16.9 ± 0.7	46.2 ± 0.6	Espada	T1	0.10 ± 0.00	19.3 ± 2.3	55.1 ± 2.1
	T2	0.04 ± 0.00	12.0 ± 0.3	45.8 ± 0.9		T2	0.05 ± 0.00	17.0 ± 0.7	55.8 ± 1.6
Yecora 70	T1	0.12 ± 0.01	15.9 ± 0.4	45.4 ± 2.7	Mace	T1	0.08 ± 0.01	16.4 ± 2.1	51.1 ± 0.7
	T2	0.05 ± 0.00	11.7 ± 0.3	43.9 ± 1.4		T2	0.04 ± 0.00	12.3 ± 1.2	56.3 ± 1.2
Magenta	T1	0.18 ± 0.00	16.9 ± 1.2	45.1 ± 2.5	Gladius	T1	0.08 ± 0.01	12.8 ± 1.5	51.5 ± 1.2
	T2	0.07 ± 0.01	12.5 ± 0.4	44.6 ± 1.4		T2	0.04 ± 0.00	13.0 ± 2.2	50.2 ± 4.1
Janz	T1	0.16 ± 0.01	16.2 ± 1.2	47.6 ± 1.7	Kukri	T1	0.08 ± 0.01	16.1 ± 2.7	49.3 ± 1.8
	T2	0.08 ± 0.01	13.1 ± 0.6	44.2 ± 2.1		T2	0.03 ± 0.00	8.9 ± 1.2	52.4 ± 0.5
Wyalkatchem	T1	0.15 ± 0.02	13.2 ± 0.5	40.4 ± 0.6	Emu Rock	T1	0.05 ± 0.01	11.9 ± 2.2	49.7 ± 1.2
	T2	0.08 ± 0.01	11.5 ± 0.3	44.3 ± 4.8		T2	0.03 ± 0.00	11.6 ± 2.0	51.9 ± 0.5
					Wyalkatchem	T1	0.07 ± 0.01	11.2 ± 1.3	45.9 ± 2.1
					(reference)	T2	0.04 ± 0.01	10.9 ± 1.9	56.0 ± 2.9
*p* value: G		<0.001	<0.001	<0.001			<0.001	<0.001	0.004
T		<0.001	<0.001	0.019			<0.001	<0.001	0.000
G × T		0.003	0.139	0.048			0.006	0.241	0.012
LSD value: G		0.023	1.662	4.154			0.013	3.152	3.835
T		0.010	0.743	1.858			0.005	1.344	1.635
G × T		0.032	NS	5.874			0.018	NS	5.424

**Table 3 plants-11-02174-t003:** Root dry weight and shoot:root ratios of 10 wheat genotypes in Batch 02 under ambient (T1) and high (T2) temperature regimes. Values presented are means ± SEM, *n* = 4.

Genotype (G)	Treatment (T)	Root Dry Weight (g plant^−1^)	Shoot:Root Ratio
Drysdale	T1	2.2 ± 0.3	8.9 ± 1.0
	T2	0.9 ± 0.0	15.9 ± 0.5
LongReach-Envoy	T1	2.0 ± 0.2	6.2 ± 0.5
	T2	1.1 ± 0.1	10.6 ± 0.2
Hartog	T1	1.7 ± 0.1	8.6 ± 0.8
	T2	0.5 ± 0.1	18.2 ± 0.7
Glossy-Huguenot	T1	3.1 ± 0.2	6.4 ± 0.5
	T2	2.1 ± 0.2	6.8 ± 0.5
RAC 875	T1	2.1 ± 0.2	10.0 ± 0.8
	T2	1.3 ± 0.1	12.4 ± 0.7
Espada	T1	2.3 ± 0.4	8.5 ± 0.5
	T2	1.5 ± 0.2	11.4 ± 1.1
Mace	T1	1.5 ± 0.2	10.8 ± 1.1
	T2	0.9 ± 0.1	13.8 ± 0.4
Gladius	T1	1.7 ± 0.3	7.8 ± 0.8
	T2	1.3 ± 0.2	10.2 ± 0.9
Kukri	T1	1.6 ± 0.3	10.5 ± 0.5
	T2	0.6 ± 0.1	15.4 ± 1.6
Emu Rock	T1	1.1 ± 0.3	12.3 ± 1.7
	T2	0.9 ± 0.1	13.3 ± 1.5
Wyalkatchem	T1	1.6 ± 0.5	8.2 ± 1.4
	T2	0.9 ± 0.2	12.5 ± 1.5
*p* value: G		<0.001	<0.001
T		<0.001	<0.001
G × T		0.540	<0.001
LSD value: G		0.462	1.946
T		0.197	0.830
G × T		NS	2.752

**Table 4 plants-11-02174-t004:** Summary of the fixed effects of the linear mixed model used to evaluate the effect of total water use per plant and leaf-to-air temperature differential (T_air_–T_leaf_) on biomass reduction% under high temperature conditions (T2).

	Degrees of Freedom (df)	Sum of Squares	Wald Statistic	Pr (Chisq)
(Intercept)	1	24197	91.640	<0.0001
(T_air_–T_leaf_)	1	16	0.061	0.806
Total water use per plant	1	11524	43.644	<0.0001
(T_air_–T_leaf_) × genotype	19	12273	46.482	0.0004
Total water use × genotype	19	13467	51.003	<0.0001
Residual (Mean Square)	<0.0001			

**Table 5 plants-11-02174-t005:** Parameters of linear regression models for the relationship between leaf temperature (T_leaf_) and air temperature (T_air_) in 20 wheat genotypes. Data are means ± SEM for the best-fit values and the goodness of fits of the regressions.

Genotype	Slope (°C °C^−1^)	T_leaf_–Intercept (°C)	R^2^	*p*
Einkorn	0.49 ± 0.15	13.73 ± 4.55	0.45	0.008
RAC 875	0.57 ± 0.12	12.44 ± 3.24	0.60	0.0002
Excalibur	0.57 ± 0.12	10.58 ± 3.36	0.63	0.0003
Espada	0.58 ± 0.14	12.57 ± 3.97	0.53	0.001
LongReach-Envoy	0.62 ± 0.14	12.57 ± 3.84	0.59	0.001
Emu Rock	0.64 ± 0.17	11.35 ± 4.84	0.49	0.002
Kukri	0.65 ± 0.16	11.15 ± 4.48	0.53	0.001
Wyalkatchem (Batch 02)	0.67 ± 0.13	10.01 ± 3.72	0.65	0.0001
Mace	0.69 ± 0.16	9.50 ± 4.40	0.56	0.001
Glossy-Huguenot	0.68 ± 0.12	9.90 ± 3.51	0.65	<0.0001
Gladius	0.74 ± 0.13	8.48 ± 3.68	0.69	<0.0001
Hartog	0.78 ± 0.13	7.51 ± 3.54	0.72	<0.0001
Drysdale	0.80 ± 0.14	6.68 ± 4.01	0.68	<0.0001
Downey	0.81 ± 0.13	4.96 ± 3.72	0.74	<0.0001
Wyalkatchem (Batch 01)	0.81 ± 0.13	3.53 ± 3.85	0.79	0.0001
Yecora 70	0.87 ± 0.19	1.64 ± 5.60	0.61	0.001
Glennson 81	0.89 ± 0.08	0.81 ± 2.26	0.92	<0.0001
Ciano 67	0.91 ± 0.19	1.26 ± 5.42	0.64	0.0004
Magenta	0.92 ± 0.12	0.34 ± 3.53	0.81	<0.0001
Sonora 64	0.93 ± 0.18	0.75 ± 5.26	0.71	0.0003
Janz	1.02 ± 0.14	−2.01 ± 4.14	0.80	<0.0001

**Table 6 plants-11-02174-t006:** Morpho-physiological criteria used for the selection of wheat genotypes.

Batch No	Character of Interest	Status	Genotype	Reference
1	Canopy Temperature Depression	High	Glennson 81	[54]
		Low	Sonora 64	[54]
	Transpiration rate or stomatal conductance (g_s_)	High	Excalibur	[88]
	Pubescence	Pubescent	Downy	[89]
		Glabrous	Einkorn *	[90]
	Leaf angle	Erectophile	Ciano 67	[91]
		Planophile	Yecora 70	[91]
	Early vigour	High	Magenta	[92]
		Low	Janz	[93,94,95]
	Adaptability to Western Australian conditions	High	Wyalkatchem	[96] Dr. D. Mullan, InterGrain, pers. Comm.
2	Carbon Isotope Discrimination (CID) or Transpiration Efficiency (TE)	Low CID or High TE	Drysdale LongReach-Envoy	[97]; Dr. D. Mullan, InterGrain, pers. Comm.; Dr. G. Rebetzke, CSIRO, pers. Comm. Dr. B. Jacobs, LongReach Plant Breeders, pers. Comm.
		High CID or Low TE	Hartog	[97]
	Glaucousness (waxiness)	Non–glaucous	Glossy–Huguenot **	[98]
		Glaucous (high)	RAC 875, Espada	[85,99]
	Grain size	High	Emu Rock	[100]
	Drought adaptability	Drought-tolerant	Mace Gladius	[101] [102]; Dr. D. Mullan, InterGrain, pers. Comm.
		Drought-susceptible	Kukri	[88,103]
	Adaptability to Western Australian conditions	High	Wyalkatchem (reference for comparisons)	[96]; Dr. D. Mullan, InterGrain, pers. Comm.

* *T. monococcum*, ** *T. turgidum*.

## Data Availability

All relevant data is contained within the article and supplementary materials. Additional data that support the findings of this study are available on request from the corresponding author.

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
