# Peer review of "Transpirational Leaf Cooling Effect Did Not Contribute Equally to Biomass Retention in Wheat Genotypes under High Temperature"

_plants, 2022, doi:10.3390/plants11162174_

Round 1
Reviewer 1 Report
In the article entitled "Transpirational Leaf Cooling Effect Did Not Contribute Equally to Biomass Retention in Wheat Genotypes under High Temperature", The authors have doe a good piece of work. I have reviewed the manuscript and found it suitable for publication.
Author Response
We highly appreciate encouraging comments from the reviewer
Reviewer 2 Report
The manuscript „Transcriptional Leaf Cooling Effect Did Not Contribute Equally to Biomass Retention in Wheat Genotypes under High Temperature” by Bramley et al. concerns the topic of heat avoidance mechanisms, especially transpirational leaf cooling in 20 morpho-physiologically diverse wheat genotypes. Heat avoidance mechanisms are much less studied than heat tolerance mechanisms, so the Authors raised an important, not yet sufficiently explored topic.
The Introduction part of the article is clearly written and it includes all important information.
The experiments were conducted with valid methodologies.
The results are clearly reported in the text. A valid discussion section in perspective to the results and literature has been presented. The conclusions are clearly formulated. I suggest, however, to emphasize the innovative aspects of the presented research and its future perspectives, including practical application.
Generally I find the article acceptable for the publication as I found only few minor editorial errors, namely:
- Line 52 – 57 references could be added
- Line 88 – 91 – references could be added
- Figure 2 – The marking (A) T1 is missing on the upper part of the figure, moreover under the upper part of the figure there are some additional dots.
- Line 297 – I guess that something is missing at the beginning of the sentences or ‘Authors’ is added by mistake
- Line 601 – ‘this’ at the beginning of the sentence is unnecessary
- Figure S1 – the quality of the figure could be improved, now it is blurred
Author Response
Point 1: Heat avoidance mechanisms are much less studied than heat tolerance mechanisms, so the Authors raised an important, not yet sufficiently explored topic. The Introduction part of the article is clearly written and it includes all important information. The experiments were conducted with valid methodologies. The results are clearly reported in the text. A valid discussion section in perspective to the results and literature has been presented. The conclusions are clearly formulated.
Response 1: We highly appreciate encouraging comments from the reviewer.
Point 2: The conclusions are clearly formulated. I suggest, however, to emphasize the innovative aspects of the presented research and its future perspectives, including practical application.
Response 2: Following points are included in the conclusion of the revised manuscript.
Point 3: Line 52 – 57 references could be added
Response 3: Following references are added in the revised manuscript (L 53-59).
- Sarkar, S.; Islam, A.A.; Barma, N.C.D.; Ahmed, J.U. Tolerance mechanisms for breeding wheat against heat stress: a review. Afr. J. Bot. 2021, 138, 262-277
- Prasad, P.V.; Bheemanahalli, R.; Jagadish, S.K. Field crops and the fear of heat stress—opportunities, challenges and future directions. Field Crops Res. 2017, 200, 114-121.
- Peguero-Pina, J.J.; Vilagrosa, A.; Alonso-Forn, D.; Ferrio, J.P.; Sancho-Knapik, D.; Gil-Pelegrín, E. Living in drylands: Functional adaptations of trees and shrubs to cope with high temperatures and water scarcity. Forests. 2020, 11(10), 1028.
- Matsumoto, J.; Muraoka, H.; Washitani, I. Ecophysiological mechanisms used by Aster kantoensis, an endangered species, to withstand high light and heat stresses of its gravelly floodplain habitat. Bot. 2000, 86(4), 777-785.
- Yadav, S.K.; Tiwari, Y.K.; Kumar, D.P.; Shanker, A.K.; Lakshmi, N.J.; Vanaja, M.; Maheswari, M. Genotypic variation in physiological traits under high temperature stress in maize. Res. 2016, 5(2), 119–126
- Jagadish, S.V.K.; Murty, M.V.R.; Quick, W.P. Rice responses to rising temperatures–challenges, perspectives and future directions. Plant Cell Environ. 2015, 38(9), 1686-1698.
Point 4: Line 88 – 91 – references could be added
Response 4: Following references are added in the revised manuscript (L90-93).
- Trnka, M.; Feng, S.; Semenov, M.A.; Olesen, J.E.; Kersebaum, K.C.; Rötter, R.P.; Semerádová, D.; Klem, K.; Huang, W.; Ruiz-Ramos, M.; Hlavinka, P.; Meitner, J.; Balek, J.; Havlík, P.; Büntgen, U. Mitigation efforts will not fully alleviate the increase in water scarcity occurrence probability in wheat-producing areas. Adv. 2019, 5(9), doi: 10.1126/sciadv.aau24
- Sadok, W.; Lopez, J.R.; Smith, K. P. Transpiration increases under high‐temperature stress: Potential mechanisms, trade‐offs and prospects for crop resilience in a warming world. Plant Cell Environ. 2021, 44(7), 2102-2116.
- Nassiri, M.; Koocheki, A.; Kamali, G.A.; Shahandeh, H. Potential impact of climate change on rainfed wheat production in Iran. Agron. Soil Sci. 2006, 52(1), 113-124.
Point 5: Figure 2 – The marking (A) T1 is missing on the upper part of the figure, moreover under the upper part of the figure there are some additional dots.
Response 5: Corrected.
Point 6: Line 297 – I guess that something is missing at the beginning of the sentences or ‘Authors’ is added by mistake
Response 6: Deleted (L.362 in the revised manuscript)
Point 7: Line 601 – ‘this’ at the beginning of the sentence is unnecessary
Response 7: Deleted (L. 688 in the revised manuscript)
Point 8: Figure S1 – the quality of the figure could be improved, now it is blurred
Response 8: The quality of the Figure S1 is improved in the revised version.
